# Development of a DAS–ELISA for Gyrovirus Homsa1 Prevalence Survey in Chickens and Wild Birds in China

**DOI:** 10.3390/vetsci10050312

**Published:** 2023-04-25

**Authors:** Shicheng Zhang, Jianhao Yang, Defang Zhou, Tianxing Yan, Gen Li, Xiaojing Hao, Qi Yang, Xiangyu Cheng, Hengyang Shi, Qing Liu, Yubao Li, Ziqiang Cheng

**Affiliations:** 1College of Veterinary Medicine, Shandong Agricultural University, Tai’an 271018, China; zscsdau@126.com (S.Z.);; 2Shanghai Veterinary Research Institute, Chinese Academy of Agricultural Sciences (CAAS), Shanghai 200241, China; 3College of Veterinary Medicine, Qingdao Agricultural University, Qingdao 266109, China; 4Qingdao Husbandry and Veterinary Institute, Qingdao 266199, China; 5Zoo Wildlife Hospital, Jinan 250032, China; 6College of Agronomy and Agricultural Engineering, Liaocheng University, Liaocheng 252059, China

**Keywords:** Gyrovirus homsa1, prevalence, DAS–ELISA, chickens, wild birds

## Abstract

**Simple Summary:**

We developed the DAS–ELISA to investigate the infection and prevalence of GyH1 in chickens and wild birds in China. We found that natural GyH1 infections were widespread in chickens and wild birds. Importantly, young chickens are more susceptible to GyH1, and local chicken species in China are genetically resistant to GyH1. In particular, we found a significantly higher GyH1–positive rate in wild birds than in chickens, implying that there may be a risk of GyH1 transmission from wild birds to chickens.

**Abstract:**

Gyrovirus homsa1 (GyH1) is an emerging pathogenic single–stranded circular DNA virus that leads to immunosuppression, aplastic anemia, and multisystem damage in chickens. However, the prevalence of GyH1 infection in chickens and wild birds remains unknown. Here, we developed a double–antibody sandwich enzyme–linked immunosorbent assay (DAS–ELISA) to investigate GyH1 infection in 8 chicken species and 25 wild bird species. A total of 2258 serum samples from chickens (*n* = 2192) in 15 provinces, and wild birds (*n* = 66) in Jinan Wildlife Hospital were collected from 2017 to 2021 in China. The GyH1–positive rates in chickens and wild birds were 9.3% (203/2192) and 22.7% (15/66), respectively. GyH1 was present in all flocks in 15 provinces. From 2017 to 2021, the positive rate ranged from 7.93% (18/227) to 10.67% (56/525), and the highest positive rate was present in 2019. Upon chicken age, the highest positive rate (25.5%) was present in young chickens (14–35 days old). Moreover, the GyH1–positive rate in broiler breeders (12.6%, 21/167) was significantly higher than that in layer chickens (8.9%, 14/157). This study shows that GyH1 has spread in chicken flocks and wild birds, and the higher GyH1–positive rate in wild birds indicates the risk of spillover from wild birds to chickens. Our study expanded the GyH1 epidemiological aspects and provided a theoretical basis for GyH1 prevention.

## 1. Introduction

Anelloviruses are a group of nonenveloped, small single–stranded circular DNA viruses ranging from 1600–3900 nt. In studies over the past 5 years, many anelloviruses have been identified in various organisms, most of which are known to infect humans [1]. Gyroviruses (GyVs), genus Gyrovirus, were assigned to the *Anelloviridae* family in 2017 [2]. Gyroviruses were first identified in human skin, blood, and subsequently in multiple hosts worldwide [3,4]. The wide distribution across multiple spaces and hosts suggests a possible pathogenic hazard. In addition, the virus is highly resistant to chemical and thermal inactivation, which allows it to persist and spread in the environment [3,5,6]. Avian species have been reported to carry many gyroviruses, including the highly prevalent chicken anemia virus (CAV), a recognized pathogen of chickens worldwide [7]. Fourteen gyrovirus species have been described over the past decade, most detected in chickens and wild migratory birds. However, these species are poorly studied, with suspected or unknown pathogenicity, zoonotic potential, and limited known host range. The pathogenic mechanisms of these GyVs remain unknown, but the genetic and structural similarities between animal and human sequences further indicate that they may be zoonotic.

Gyrovirus homsa1 (GyH1), originally named Gyrovirus 3 (GyV3), was a member of the Gyrovirus [4]. The virus was initially found in the feces of a child with acute diarrhea [8]. Subsequently, the virus was detected in commercially traded poultry, healthy Brazilian wild birds, chickens with transmissible viral proventriculitis (TVP), and various mammals, suggesting that GyH1 may be the causative virus [9,10,11,12,13]. More importantly, pathogenicity experiments have shown that GyH1 causes aplastic anemia, immunosuppression, and severe multisystem damage in chickens and mice [14,15]. Furthermore, molecular epidemiology investigation showed that the GyH1 infection rate in broiler chickens with TVP was 12.5% from 2013 to 2017, suggesting that GyH1 is highly associated with TVP [12]. However, the prevalence of GyH1 in field chickens and wild birds is unclear. Rapid, efficient, and high–throughput assays are urgently needed to detect GyH1 infection. Here, we developed a double–antibody sandwich enzyme–linked immunosorbent assay (DAS–ELISA) to investigate the prevalence of GyH1 in chickens and wild birds in China.

## 2. Materials and Methods

### 2.1. Virus Strain

GyH1 strain SDAU–1 (GenBank accession number MG366592) is maintained in the Laboratory of Molecular Pathology, Shandong Agricultural University, China.

### 2.2. Sample Collection

A total of 2192 chicken serum samples were collected between 2017 and 2021, with sample information including temporal and spatial distribution, species, and age. The spatial distribution of the sample covers 15 provinces in China, including Jiangsu (JS), Liaoning (LN), Guangdong (GD), Guangxi (GX), Gansu (GS), Anhui (AH), Zhejiang (ZJ), Shanxi (SX), Jilin (JL), Heilongjiang (HLJ), Fujian (FJ), Sichuan (SC), Henan (HN), Hunan (HN), and Shandong (SD). The samples cover eight chicken species, including Ma chicken, Sanhuang chicken, Hy–line Brown, Black chicken, Rose 308, Bantam, White Plymouth Rock, and Hubbard. Likewise, we randomly divided the chickens into five stages by age, including 1–14 days, 14–35 days, 35–98 days, 98–189 days, and over 189 days.

During the collection of the samples, the flocks examined were apparently healthy. To have an overall fair and representative description of the situation in the field, according to the principle of random sampling, we collected the sera of the selected chickens. Blood samples were drawn by venipuncture into dry vacuum tubes and cooled to 4 °C overnight. The sera were harvested by centrifugation at 700× *g* for 5 min. Additionally, sera were stored at −20 °C until they were assayed, which was performed in triplicate.

Shandong’s geographic location makes it an essential migratory habitat for wintering and stopovers for migratory birds crossing East Asia and Australia [16]. We collected a total of 66 serum samples from rescued birds at the Zoo Wildlife Hospital in Jinan, Shandong Province, China. The attending veterinarian determined the reasons for admission of the birds and recorded various data on each bird sampled, including date, the reason for admission, and clinical presentation. We took blood samples by venipuncture from birds. The samples were assayed using DAS–ELISA.

### 2.3. Ethics Statement

The rescue and handling of the wild birds in this study were carried out following the Wildlife Protection Law of the People’s Republic of China. In addition, all our animal experimental procedures were approved by the Ethics and Morality Committee of Shandong Agricultural University (permit No. SDAU–20170729).

### 2.4. Expression and Purification of the VP1 Protein

Viral DNA was extracted from GyH1–infected tissue using a commercial kit (TIANGEN, Beijing, China) according to the manufacturer’s instructions. Optimized expression of the synthetic gene in *E. coli* was based on the viral protein 1 (VP1) sequence. Using *Sac I* and *Not I* restriction sites, the VP1 sequence was encoded by cloning into the *E. coli* expression vector pET32a (+) and expressed in BL21 competent cells. Recombinant protein expression and purification of nickel–chelated agarose were performed under denaturing conditions. Proteins were dialyzed in 0.01 M phosphate–buffered saline (PBS, TIANGEN, Beijing, China, pH 7.2), and SDS–PAGE was used to detect recombinant VP1 protein. Western blot analysis was then performed using anti–GyH1–positive chicken serum.

### 2.5. Production of Monoclonal Antibodies against the VP1 Protein of GyH1

To generate hybridomas that secrete anti–GyH1–specific monoclonal antibodies (mAbs), we emulsified purified recombinant VP1 antigen in complete Freund’s adjuvant (Sigma, Aldrich, USA) for the initial immunization. Freund’s incomplete adjuvant (Sigma Aldrich, Burlington, MA, USA) was applied for the subsequent three immunization doses (approximately 80 μg recombinant VP1 protein content/immunization). After repeated immunization of mice, an antiserum against the GyH1–VP1 protein was successfully produced. After the last immunization, spleen cells were collected from mice and fused with SP2/0 cells. The fused cells were cultured in HAT medium (Sigma, Ronkonkoma, NY, USA) containing hypoxanthine (H), aminopterin (A) and thymidine deoxyriboside (T) for 14 days. Screened hybridoma cells were then cultured in Dulbecco’s modified Eagle’s medium (DMEM) containing 15% fetal bovine serum (FBS) (Gibco, Gaithersburg, MD, USA). Culture supernatants of hybridomas were examined by indirect ELISA, coated with purified VP1 protein, and positive wells were subjected to restriction dilution and further amplification. To obtain mAbs against GyH1, we inoculated antibody–secreting cell lines into mice to induce large amounts of antibody–containing ascites. Immunoglobulin G (IgG) in the ascites was purified with protein A Sepharose (GE Healthcare, Chicago, IL, USA). The purified mAbs (2B5, 2F2, 3F5, 3H1 and 3G7) were then analyzed for reactivity and specificity by Western blot. 

### 2.6. Development of DAS–ELISA

Different concentrations of mAb were used for capture (0.1875, 0.375, 0.75, 1.5, 3, 6, 12, and 24 μg/mL) and detection (0.1875, 0.375, 0.75, 1.5, 3, 6, 12, and 24 μg/mL), with a tessellation test used to determine the optimum concentration for each. The DAS–ELISA was performed using the following procedure. Briefly, 96–well plates were coated with capture mAb diluted in 0.01 M carbonate buffer saline (CBS, 100 µL/well) and incubated overnight at 4 °C. The plates were washed three times with phosphate–buffered saline tween (PBST) and then incubated with 4% skimmed milk (200 µL/well) for 1 h at 37 °C. After three washes with PBST, positive serum from GyH1–infected chickens (100 µL/well) diluted 1:4 in PBS was added and incubated for 1 h at 37 °C. Afterwards, the plate was washed three times, and 100 μL of diluted horseradish peroxidase (HRP)–conjugated assay mAb was added to each well. After 1 h at 37 °C, 3,3′,5,5′–Tetramethylbenzidine (TMB) substrate solution (100 µL/well) was added to each well and incubated for 15 min at 37 °C in the dark. The reaction was stopped by adding 50 µL of 2 M H_2_SO_4_, and the optical density (OD) value was read at 450 nm using an automated ELISA plate reader (Thermo Scientific™, Multiskan SkyHigh, Waltham, MA, USA).

### 2.7. Determination of DAS–ELISA Cut–Off Value

Under the optimum conditions determined, 32 negative serum samples from specific pathogen–free (SPF) chickens were tested by the established DAS–ELISA to determine the cut–off value. The cut–off values defining viral positivity were calculated according to the following formula: positive/negative cut–off value = mean of negative samples + 3 standard deviations (SD). 

### 2.8. Sensitivity, Specificity and Repeatability of the DAS–ELISA

To assess the sensitivity of DAS–ELISA in detecting GyH1, we added 100 μL GyH1–positive serum and GyH1–negative serum in serial quadruplicate dilutions with PBS to microtiter plates. Positive sera were collected from pathogenicity assays and validated by Western blotting. In addition, positive serum was diluted at 1:1, 1:4, 1:16, 1:64, 1:256, 1:1024, 1:4096, and 1:16,384. Samples were then assayed according to the optimization of DAS–ELISA. A standard curve of OD_450_ values versus serum dilution was plotted to determine the detection limit and sensitivity of the DAS–ELISA.

The specificity of the DAS–ELISA was assessed by testing a positive control (GyH1) for other avian pathogens, including avian leukosis virus subgroup J virus (ALV–J), reticuloendotheliosis virus (REV), Marek’s disease virus (MDV), infectious laryngotracheitis virus (ILTV), infectious bursal disease virus (IBDV), egg drop syndrome virus (EDSV), infectious bronchitis virus (IBV), avian influenza virus (H9N2), Newcastle disease virus (NDV), and CAV. Based on experimental results, we used the ROC algorithm and statistical analysis to assess specificity. 

Here, 6 technical replicates of 6 samples (3 positive controls and 3 negative controls) were performed in 96–well plates coated with capture mAbs to verify the intra–batch reproducibility of the DAS–ELISA. Subsequently, we assayed 6 samples in 6 different batches of 96–well plates coated with capture mAbs to verify the inter–batch reproducibility. The intra– and interassay coefficient of variation (CV) was calculated using the following formula: CV = standard deviation (SD)/mean OD450 value of the sample × 100%. The reproducibility and stability of the method were evaluated based on the results. All of these tests were repeated at least 3 times.

### 2.9. Comparison of DAS–ELISA and qPCR

Fifty–four blood samples from artificially infected GyH1 were tested by DAS–ELISA and quantitative real–time PCR (qPCR). The qPCR primer sequences and amplification procedures required were as described by Yuan et al. Ct values less than 30 were regarded as GyH1–positive [14]. We subsequently compared the results of the DAS–ELISA with those of qPCR to assess the accuracy of the DAS–ELISA.

### 2.10. Statistical Analysis

GyH1–positive rates were calculated by dividing the number of positive animals by the total number of animals tested, using a two–sided exact binomial test with a 95% confidence interval (CI). All data are reported as mean ± standard error of the mean. SPSS software (version 25.0; IBM, NY, USA) was used to determine statistically significant differences using a two–tailed unpaired Student’s *t*-test. Results with *p* values less than 0.05 were considered statistically significant. Bar graphs were created using GraphPad Prism (version 8.1; San Diego, CA, USA).

## 3. Results

### 3.1. Development of DAS–ELISA

We optimized the reaction conditions for the DAS–ELISA to obtain maximum results. Initially, we determined that 3 µg/mL capture antibody 2F2 and 6 µg/mL detection antibody 3H1 were the optimal pair for DAS–ELISA (Figure 1A). Briefly, 96–well plates were coated with 0.3 µg/well of purified mAb 2F2 (3 µg/mL), which was diluted in sodium carbonate buffer (pH 9.4). Plates were incubated overnight at 4 °C. Subsequently, 4% skim milk dissolved in PBST was incubated for 1 h at 37 °C (Figure 1B,C). Next, samples to be tested were diluted to 1:4 in PBS and incubated for 1 h at 37 °C (Figure 1D). The detection antibody was then diluted to 6 µg/mL with CBS and incubated at 37 °C for 1 h (Figure 1E). TMB chromogenic solution was added and incubated for 15 min at 37 °C in the dark (Figure 1F). The reaction was terminated with 2 M H_2_SO_4_ and read in a microplate reader.

The mean and standard deviation of the 32 SPF negative sera were 0.193 and 0.019, respectively, so the cut–off value was set to 0.251 (0.193 + 3 × 0.019) (Figure 1G). The detection limit of the DAS–ELISA was 1:256, and the sensitivity by ROC curve analysis was 93.14% (Figure 1H). No cross–reactivity was detected by DAS–ELISA using positive sera between ALV, REV, MDV, ILTV, NDV, EDSV, IBDV, IBV, H9 subtype AIV, CAV, and GyH1, with OD values ranging from 0.127 to 0.209 (Figure 1I). The CVs for the six control sera tested by ELISA were less than 10%. The intra–batch CVs ranged from 3.62% to 9.37%, while the inter–batch CVs ranged from 2.25% to 7.2%, indicating reproducible results (Figure 1J). Moreover, comparative experiments showed a 94.14% compliance rate between DAS–ELISA and qPCR, indicating the high accuracy and clinical application of DAS–ELISA (Table 1).

### 3.2. Monitoring the Prevalence of GyH1 in Chickens and Wild Birds

We used the developed DAS–ELISA to examine chicken (*n* = 2192) and wild bird *(n* = 66) serum samples to investigate the prevalence of GyH1 infection. The GyH1–positive rate in chickens was 9.3% (203/2192) and ranged from 3.5% (5/144) to 13.7% (19/139) in 15 provinces (Figure 2A). Upon longitudinal investigation, the GyH1–positive rate ranged from 7.93% (18/227) to 10.67% (56/525) between 2017 and 2021, with the highest positive rate occurring in 2019 (Figure 2B). Of the eight chicken species, the GyH1–positive rate ranged from 4.5% (7/157) to 12.3% (19/155), with Ma chickens (4.5%) and San-huang chickens (5.3%) having lower positive rates (Figure 2C). In addition, the GyH1–positive rates in broiler breeders and layer chickens were 12.6% (21/167) and 8.9% (14/157), respectively (Figure 2D). Subsequently, we found that the GyH1–positive rate was negatively correlated with age and showed significantly higher infection rate (25.5%) at the second age stage (14~35 days) than at the other age stages (4.6–14.5%) (Figure 2E). Sixty–six serum samples were collected from twenty–five rescued wild birds. According to the admission forms and clinical records sent from the Wildlife Hospital, the birds in this study were admitted for three main reasons: trauma (10/66; 15.2%), clinical signs/disease (35/66; 53.0%), and animal attack (8/66; 12.1%). In addition to unspecified reasons (7/66; 10.6%), birds were admitted for other miscellaneous reasons (6/66; 9.1%), including land clearing, inability to fly, and being trapped. Using DAS–ELISA, the GyH1–positive rate in wild birds was 22.7% (15/66) (Table 2).

## 4. Discussion

Gyrovirus is a single–stranded, cyclic, capsid–free virus. Studies have shown that gyrovirus can survive in low pH environments, such as gastric acid, suggesting that the virus may be highly resistant to chemical and thermal inactivation, which allows it to persist and spread in the environment [6]. Ubiquitous gyroviruses make chickens in confinement vulnerable to the virus [17]. Moreover, the same host will likely be infected with multiple gyroviruses [18]. Co–infection of GyH1 with other gyroviruses, e.g., gyrovirus galga1 (GyG1) and CAV, occurs frequently. Importantly, GyH1 co–infection with CAV leads to higher mortality and more widespread lymphocyte apoptosis [19], which increases the risk posed and makes prevention and control more difficult. Specifically, epidemiology investigation suggests that GyH1 is widespread in China, which is consistent with our findings [17,18,20]. Thus, an accurate diagnosis of GyH1 is particularly important.

Recombination in different viral genomes or within viral genomes is thought to be a major driver of viral evolution [21]. Previous studies have not found the recombination of GyH1 with other viruses [18]. Furthermore, GyH1 consists of three major proteins, the capsid protein VP1, the backbone protein VP2, and the apoptosis protein VP3. The VP1 protein plays an important role in viral virulence, replication, and infection of host cells [14,19]. For this reason, we prepared a series of anti–GyH1 monoclonal antibodies based on a highly antigenic and conserved fragment of the VP1 protein. Afterwards, we screened five highly effective monoclonal antibodies by immunoblot identification. Antibody pairing experiments showed a good binding response between 2F2 and 3H1. Subsequently, we developed a DAS–ELISA for the detection of GyH1. We optimized the reaction conditions and finalized the cut–off value for the DAS–ELISA. Furthermore, the developed DAS–ELISA does not cross–react with other chicken viruses except GyH1. More importantly, we determined that DAS–ELISA had a sensitivity of 93.14% and an accuracy of 94.14%, indicating that DAS–ELISA is of high clinical application.

We selected standardized chicken farms in 15 provinces with a size of 500,000 chickens to investigate the spatial distribution characteristics of GyH1 infection. We used SPSS software to perform statistical tests and compare differences. There was no significant difference in the number of positive chickens in most provinces, demonstrating that natural GyH1 is widespread in chickens. Changes and interactions between pathogens, hosts, and the environment often result in changes in disease. We explored the possible transmission patterns of GyH1 through the temporal distribution characteristics of its emergence in China to formulate prevention and control measures. We found that the highest positive rate of GyH1 occurred in 2019, which may be because after the pathogenicity of GyH1 attracted widespread attention, breeders stepped up preventive measures to reduce the infection rate [14]. Furthermore, the molecular epidemiological and seroepidemiological results further confirm our conclusions [18,20].

Surprisingly, we detected significant differences between chicken species and positive rates of GyH1. In contrast, the positive rate of Ma and Sanhuang chickens was significantly lower than that of other chicken species, which may be due to the solid genetic disease resistance of native chickens after a long period of natural selection and evolution [22,23,24]. More intensive research is needed to further verify this conjecture. Through natural selection and evolution over time, many avian species have become hotbeds for a wide range of viruses [25,26,27]. Bats carry large numbers of coronaviruses but do not themselves develop them. In a study of the origin of novel coronaviruses, researchers found that coronavirus virulence, infection, and transmission were enhanced during the passage of coronaviruses from bats to civets [28,29,30].

Similarly, we observed a significant negative correlation between age and GyH1–positive rates. The highest positive rate present in chickens between 14 and 35 days of age indicates that GyH1 is self–limited. In our previous in vivo experiment, we noticed that infection with GyH1 in the early life of chickens triggers massive apoptosis of hematoblasts, promyeloblasts, and lymphocytes in the bone marrow, resulting in a rapid proliferation of GyH1 in a short period and a rapid increase in viral titers in the blood [14,19]. Subsequently, virus load in the blood decreases with age due to lymphocyte and promyelocyte regeneration and hematopoietic cell activity recovery. In particular, GyH1 mono–infection experiments showed that GyH1 virus shedding began at 14 days post–infection, peaked at 21 days post–infection, and began to decrease at 28 days post–infection. Additionally, immune organ dynamic monitoring experiments and erythrocyte testing experiments have further validated our results [14,15,19].

In transmission, birds are natural reservoirs for viral genes and breeding grounds for viruses that can be transmitted cross–species. These warm–blooded vertebrates exhibit a high degree of species diversity, roosting and migratory behavior, the ability to fly long distances, and possess a very versatile immune system. These are ideal qualities for asymptomatic shedding, transmitting, and mixing of various viruses to develop novel mutant, recombinant, or reassortant DNA/RNA viruses [31,32,33]. Wild birds are under constant epidemiological observation as they are natural reservoirs of ever–increasing zoonotic pathogens and have a significant impact on poultry health [34,35,36]. This epidemiological investigation demonstrates that natural GyH1 infection is widespread in wild migratory birds, although the source and manner of infection in wild migratory birds remain unclear. GyVs have been detected in many migratory birds [37,38]. The naturally high GyH1 infection in wild migratory birds indicates the increasing risk of GyH1 spillover from wild migratory birds to chickens. In the future, we will expand the scope of GyH1 epidemiological research to understand the source and host range of GyH1.

## 5. Conclusions

In conclusion, a highly sensitive and specific DAS–ELISA was developed to investigate the prevalence of GyH1 infection in chickens and wild birds. Natural GyH1 infection was widespread in chickens and wild birds, and GyH1 is prone to infect young chicks. The poultry industry should pay significant attention to the GyH1 spillover from wild migratory birds to chickens.

## Figures and Tables

**Figure 1 vetsci-10-00312-f001:**
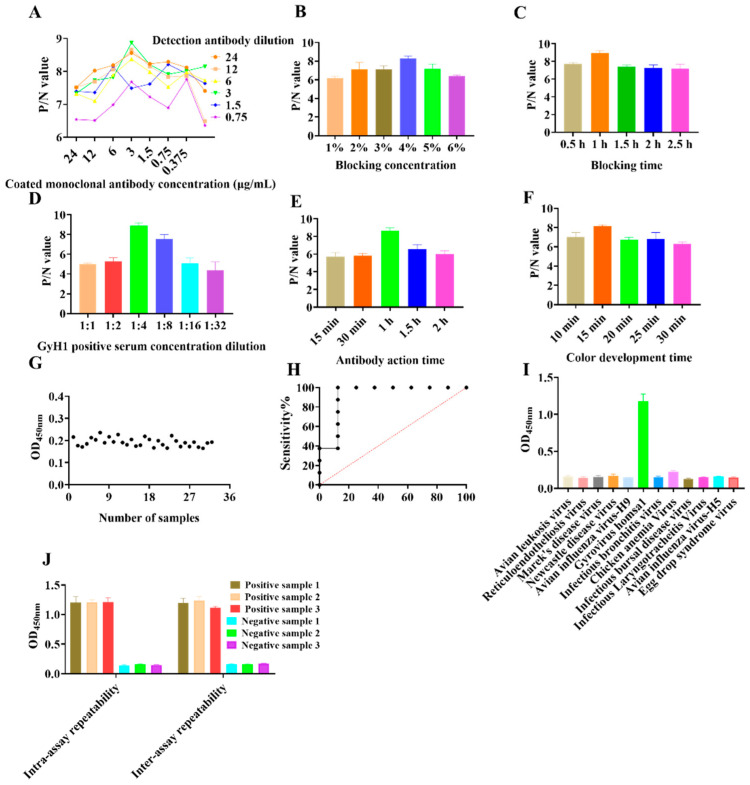
Development of the DAS–ELISA. (**A**) Determining the optimal concentration of capture and detection antibodies. (**B**) Optimal concentration of blocking incubation. (**C**) Optimal incubation time for closure. (**D**) Optimal antigen dilution. (**E**) Optimal incubation time for detection antibody. (**F**) Optimal incubation time for color development. (**G**) Determination of DAS–ELISA cut–off values. (**H**) Sensitivity test for DAS–ELISA. (**I**) DAS–ELISA cross–reactivity analysis. (**J**) Repeatability testing of the DAS–ELISA. Bar and line graphs were created using GraphPad Prism (version 8.1; San Diego, CA, USA). Data are presented as mean ± SEM (*n* = 3) (Student’s *t*-tests).

**Figure 2 vetsci-10-00312-f002:**
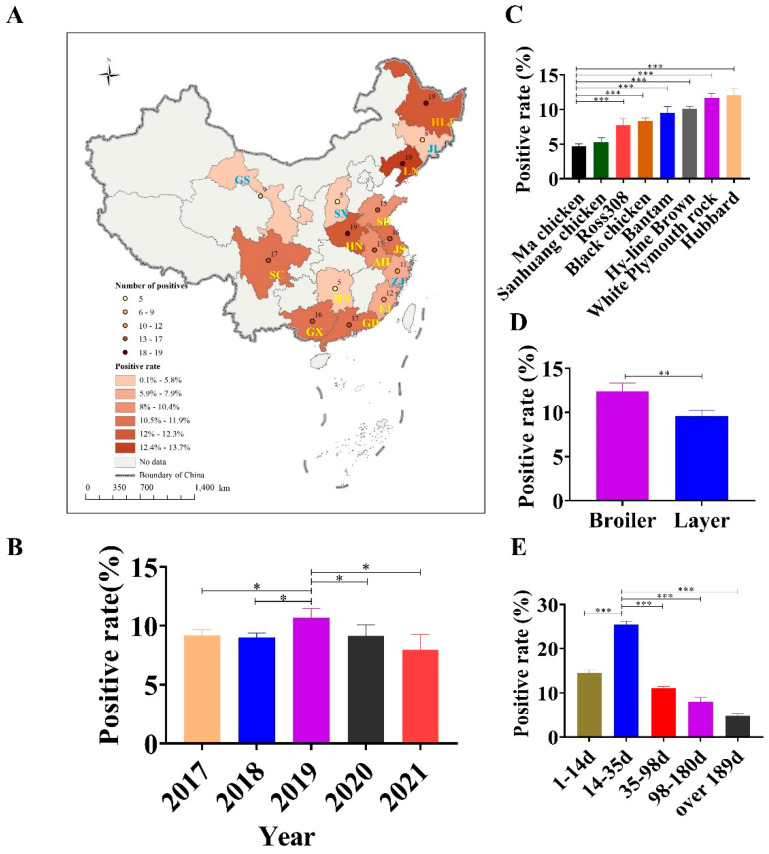
Prevalence survey of GyH1 infection in chickens. (**A**) Geographical distribution of the positive rate and number of positive samples in 15 provinces in China. (**B**) GyH1–positive rates from 2017 to 2021. (**C**) GyH1–positive rates in eight chicken species. (D) Comparison of GyH1–positive rates in layer and broiler breeders. (**E**) GyH1–positive rates in different ages. Different colors indicate the positive rate for additional years, species, and ages, respectively. Data are presented as mean ± SEM (*n* = 3). * *p* < 0.05; ** *p* < 0.01; *** *p* < 0.001 compared with the control (Student’s *t*-tests). SPSS software (version 25.0; IBM, USA) was used to determine statistically significant differences using a two–tailed unpaired Student’s *t*-test.

**Table 1 vetsci-10-00312-t001:** Comparison of the results of DAS–ELISA and qPCR.

		DAS–ELISA		
		Positive	Negative	Total
qPCR	Positive	47	6	53
	Negative	0	1	1
	Total	47	7	54

**Table 2 vetsci-10-00312-t002:** GyH1 infection in wild birds.

Species	No. of Samples	No. of Positive
Magpie	7	2
Wild Turkey	3	1
Swan	2	0
Grey Heron	1	1
Black Swan	2	0
Turtle Dove	1	1
Lovebird	2	1
Pheasant	4	0
Common Kestrel	6	2
Little Bittern	1	1
Maroon–bellied Conure	1	1
Bar–headed Goose	2	0
Ruddy Shelduck	1	1
Blue and Gold Macaw	2	0
Mute Swan	1	0
Alexandrine Parakeet	2	0
Short–eared Owl	1	0
Sun Parakeet	1	0
Night Heron	6	3
Swan Goose	2	0
Mallard	2	0
Kim–Jaggy	5	1
Cuckoo	4	0
Linnaeus	3	0
Bean goose	4	0
Total	66	15

## Data Availability

The data presented in this study are available on request from the corresponding author.

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
