# Peer review of "Development of a DAS–ELISA for Gyrovirus Homsa1 Prevalence Survey in Chickens and Wild Birds in China"

_vetsci, 2023, doi:10.3390/vetsci10050312_

Round 1
Reviewer 1 Report
In this paper, the authors report a double-antibody sandwich ELISA (DAS-ELISA) established to investigate GyH1 infection in eight chicken 18 species and 25 wild bird species. In general, the paper is interesting study, although the discussion is not exhaustive and is based on only very few articles. Please review the literature more deeply to compare your results with those of other authors.
Author Response
Point 1: In general, the paper is interesting study, although the discussion is not exhaustive and is based on only very few articles. Please review the literature more deeply to compare your results with those of other authors.
Response 1: Your suggestions have been so valuable to us. Thank you very much for your kind advice. We have reviewed the manuscript and the articles of other authors in the manuscript in depth, comparing our results more systematically with those of other authors. The additions have been highlighted in red in the manuscript.

Reviewer 2 Report
The reviewed paper describes the development a highly sensitive and specific DAS-ELISA and investigation the prevalence of GyH1 infection in chickens and wild birds. A series of anti-GyH1 monoclonal antibodies based on a genic fragment of the VP1 protein have been obtained, HRP-conjugated mAb was prepared and DAS-ELISA was development. Sensitivity and repeatability of the DAS-ELISA was assessed. DAS-ELISA does not cross-react with other chicken viruses except GyH. This assay will be useful for poultry workers.
Some notes to the text are listed below:
1) Some abbreviations are not deciphered at the first mention, for example: CBS – (line 129) PBST– (line 130).
2) Line 131: It is not clue, the VP1 protein or the positive serum is the positive control?
3) Line 130 and line 179: In the first case, blocking with skimmed milk is performed after mAb, and in the second case, after samples to be tested.
4) Table 1 is unclear
5) The caption under Figure 1A should be: “Coated monoclonal antibody concentration µg/ml”
Author Response
Response to Reviewer 2 Comments
Point 1: Some abbreviations are not deciphered at the first mention, for example: CBS – (line 129) PBST–(line 130).
Response 1: Thank you very much for your patience in reviewing this article. Your suggestions were very helpful to us. We have checked the full text and deciphered all abbreviations where possible. We have highlighted the deciphered abbreviations in the manuscript by marking them in red.
Point 2: Line 131: It is not clue, the VP1 protein or the positive serum is the positive control?
Response 2: We are very grateful for your kind advice. We have amended "positive control" to "positive serum from infected chickens" and highlighted it in the manuscript. We have previously experimented with VP1 protein, but the reaction was not as good as with positive serum.
Point 3: Line 130 and line 179: In the first case, blocking with skimmed milk is performed after mAb, and in the second case, after samples to be tested.
Response 3: We are very grateful for your kind advice. We are very sorry that our oversight has caused you inconvenience. We incubated the antibodies after the skim milk was closed. We have reformatted the figures and corrected them in line 179.
Point 4: Table 1 is unclear.
Response 4: We are very sorry for the inconvenience caused. Table 1 shows the 54 samples detected as suspicious using qPCR and DAS-ELISA. Fifty-three positives and one negative were detected by qPCR, while forty-seven positives and seven negatives were detected using DAS-ELISA. The results indicate that the DAS-ELISA has high accuracy.
Point 5: The caption under Figure 1A should be: “Coated monoclonal antibody concentration µg/ml”.
Response 5: We are very grateful for your kind advice. Your suggestion is very correct. We have corrected the caption under Figure 1A to "Coated monoclonal antibody concentration µg/ml" as you suggested.

Reviewer 3 Report
The manuscript by Zhang et al entitled, “Development of a DAS-ELISA for Gyrovirus homsa1 prevalence survey in chickens and wild birds in China”, reports development of an ELISA test for the VP1 protein of the Gyrovirus homsa1 and its use as a detection tool in serum samples from domestic and wild bird samples. The findings indicate that GyH1 is widespread among domestic and wild birds, although the virus is more prevalent in commercial chicken breeds and those between 14-35 days of age. This manuscript provides a diagnostic tool for GyH1. In the future, it would be important to further investigate the clinical significance/relevance of this virus in poultry and bird health.
Major comments
Overall:
Is the variation in VP1 known? Will the DAS-ELISA be able to detect variants in the VP1 protein? How would variations in VP1 impact the DAS-ELISA results?
Overall:
Another manuscript on GyH1 has been published in 2022 and written by some of the authors in this submission. Please see the citation below. However, this has not been cited. Can you provide a rationale? The figures are very similar to each other.
- Zhang, S., Yuan, S., Yan, T. et al. Serological investigation of Gyrovirus homsa1 infections in chickens in China. BMC Vet Res 18, 231 (2022). https://doi.org/10.1186/s12917-022-03334-0
Figure 1 B: Can you explain why the P/N ratio for 1:1 and 1:2 is lower than that of 1:4? Do you suspect factors in the serum that inhibiting antibody binding?
Figure 2: Indicate number of samples per category.
Figure 1 and 2: Please indicate statistical tests used where appropriate.
Line 68-70: Please provide a rationale of using sera to screen for GyH1 VP1 protein. What other studies have shown that this sample is most appropriate for GyH1?
Lines 122-123: Please show the data for testing reactivity and specificity of the monoclonal antibodies.
Line 152-153: Please also specify positive controls from other viruses that were used. Are these purified proteins as well?
Line 167-172: What is the detection limit of the qPCR assay? The Ct value of 30 seems to be a low number for a cut-off. Do you have a sense of how many copies Ct=30 is?
Line 175-185: Please indicate clearly that you are testing for the presence of VP1 protein in chicken sera since usually, sera is used to test for the presence of antibodies against a pathogen.
Minor comments
Lines 80-81: Can you specify how the chickens were selected? Were the serum samples collected from apparently healthy flocks or those with clinical disease?
Line 113: The antiserum referred here is not monoclonal yet.
Line 129: Please define CBS.
Line 131: Please specify what the positive was. Is it the purified VP1 protein?
Line 235: “infestation” to “infection”
Line 245-247, Figure 1A: Please indicate the statistical test used to compare prevalence. If not done, please drop the adjective “statistical” since its use must be exclusively for findings supported by statistical testing.
Line 268: “Transmmition” to “transmition”
Author Response
Point 1: Is the variation in VP1 known? Will the DAS-ELISA be able to detect variants in the VP1 protein? How would variations in VP1 impact the DAS-ELISA results?
Response 1: Thank you very much for your patience in reviewing this article. Your comments have been very valuable to us.
The variation in VP1 is known. The GyH1 isolates reported in several papers have only relatively lower nucleotide and amino acid identity than the GyH1 isolates previously reported for the first time in children's feces, indicating that the GyH1 variant is known. We constructed the prokaryotic expression vector based on the conserved region of the VP1 protein. The induced monoclonal antibodies only bind the antigenic epitopes exposed in the conserved region of VP1 and cannot bind the VP1 highly variable region. Therefore, the developed DAS-ELISA was unable to detect variant VP1 proteins. Concomitantly, VP1 variants are unable to affect DAS-ELISA results.
Next, we will construct an expression vector for the highly variable region of VP1 protein and establish indirect and double antibody sandwich ELISA methods. In addition, we will use a combined nucleic acid assay, an indirect ELISA, and a double sandwich antibody ELISA assay to conduct a systematic epidemiological survey to better identify the prevalent GyH1 genotype in chickens in China.
Point 2: Another manuscript on GyH1 has been published in 2022 and written by some of the authors in this submission. Please see the citation below. However, this has not been cited. Can you provide a rationale? The figures are very similar to each other.
- Zhang, S., Yuan, S., Yan, T., et al. Serological investigation of Gyrovirus homsa1 infections in chickens in China. BMC Vet Res 18, 231 (2022). https://doi.org/10.1186/s12917-022-03334-0
Response 2: We are very grateful for your kind advice. This article, “Serological investigation of Gyrovirus homsa1 infections in chickens in China”, is also written by me. The reason I did not cite this article is that I was afraid I would self-cite it and cause unnecessary trouble for veterinary science. In the manuscript, we investigated the antigenic prevalence of GyH1 in chickens in China. The results detected are a further argument for the article mentioned above. The results provide further evidence of the widespread presence of GyH1 in chickens in China and the differences in age resistance and susceptibility. Furthermore, the results of the following article are consistent with our results and indirectly validate our results. We have cited this literature in the appropriate places.
-- Yan, T., Zhao, M., Sun, Y., Zhang, S., Zhang, X., Liu, Q., Li, Y., & Cheng, Z. (2023). Molecular evolution analysis of three species gyroviruses in China from 2018 to 2019. Virus Research, 326, 199058. https://doi.org/10.1016/j.virusres.2023.199058
Point 3: Figure 1 B: Can you explain why the P/N ratio for 1:1 and 1:2 is lower than that of 1:4? Do you suspect factors in the serum that inhibiting antibody binding?
Response 3: You have raised a very good question. We are very grateful for your kind advice. The P/N ratio of 1:1 or 1:2 diluted serum is lower than the P/N ratio of 1:4 diluted serum. We suspect that the presence of certain factors in the serum (e.g. chicken anti-GyH1 antibody, complement, and coagulation factors) may inhibit antibody binding at high concentrations, or that the concentration of residual VP1 protein in the serum is too high to mask the original antigenic epitope. To test our hypothesis, we performed tests with GyH1-infected MDCC-MSB1 cell lysates, all other conditions being equal. The results showed that the P/N ratio at lower dilutions was significantly lower than that at 1:4 dilution, suggesting that the high protein concentration may have partially masked the antigenic epitope.
Point 4: Figure 2: Indicate number of samples per category.
Response 4:
We are very grateful for your kind advice. We have indicated the sample numbers for the category in the table below. As most of the samples tested were collected by chicken farm workers, the information on some samples was not perfect.
Table 1 Samples are grouped by year collected.
|
Year groups |
Sample numbers |
|
2017 |
403 |
|
2018 |
501 |
|
2019 |
525 |
|
2020 |
536 |
|
2021 |
227 |
Table 2 Samples are grouped by species collected.
|
Species |
Sample numbers |
|
Ma chickens |
157 |
|
Sanhuang chickens |
152 |
|
Ross308 |
147 |
|
Black chicken |
151 |
|
Bantam |
155 |
|
Hy-line Brown |
155 |
|
White Plymouth rock |
154 |
|
Hubbard |
155 |
|
Not indicated |
966 |
Table 3 Samples are grouped by use.
|
Uses groups |
Sample numbers |
|
Broiler breeders |
167 |
|
layers |
157 |
|
Not indicated |
1868 |
Table 4 The sample is grouped by age stage.
|
Age groups |
Sample numbers |
|
1-14d |
159 |
|
14-35d |
145 |
|
35-98d |
148 |
|
98-180d |
147 |
|
Over 180d |
151 |
|
Not indicated |
1442 |
|
Province |
Sample numbers |
|
Guangdong |
144 |
|
Zhejiang |
139 |
|
Guangxi |
137 |
|
Anhui |
144 |
|
Shanxi |
129 |
|
Hunan |
144 |
|
Jiangsu |
135 |
|
Jilin |
155 |
|
Liaoning |
139 |
|
Heilongjiang |
155 |
|
Fujian |
155 |
|
Sichuan |
151 |
|
Shandong |
155 |
|
Gansu |
155 |
|
Henan |
155 |
Table 5 The sample is grouped by geographical distribution.
Point 5: Figure 1 and 2: Please indicate statistical tests used where appropriate
Response 5: We are very grateful for your kind advice. We have indicated statistical tests where appropriate. SPSS software (version 25.0; IBM, USA) was used for statistical tests using a two-tailed unpaired Student's t-test. Indicated statistical tests have been highlighted in red text.
Point 6: Line 68-70: Please provide a rationale of using sera to screen for GyH1 VP1 protein. What other studies have shown that this sample is most appropriate for GyH1?
Response 6: This is a very good question. Thank you very much for your friendly advice. In a previous pre-experiment, we detected GyH1 by PCR in the sera of GyH1-infected chickens. Subsequently, we speculated that the shed viral capsid protein (VP1 protein) might be present in the sera of GyH1-infected chickens. We then performed SDS-PAGE and immunoblotting experiments on sera to confirm this suspicion. The results showed that shed VP1 protein was indeed present in the sera of GyH1-infected chickens. In these articles, "Yuan, S., Yan, T., Huang, L., Hao, X., Zhao, M., Zhang, S., Zhou, D., & Cheng, Z. (2021). Cross-species pathogenicity of gyrovirus 3 in experimentally infected chickens and mice. Veterinary microbiology, 261, 109191.", and “Yang, M., Yang, Q., Bi, X., Shi, H., Yang, J., Cheng, X., Yan, T., Zhang, H., & Cheng, Z. (2023). The Synergy of Chicken Anemia Virus and Gyrovirus Homsa 1 in Chickens. Viruses, 15(2), 515.” illustrating the detectability of GyH1 in serum, which further confirms our conclusion. Additionally, other articles have reported the detection of GyH1 in human and mammalian faeces, in chicken meat sold in supermarkets, and in the serum of seabirds.
Point 7: Lines 122-123: Please show the data for testing reactivity and specificity of the monoclonal antibodies.
Response 7:
Figure S1 Reactivity and specificity of five monoclonal antibodies by Western blotting. Lane M, protein marker; Lane N, 72 h MDCC-MSB1 cell lysates; Lane 2B5, 2F2, 3F5, 3H1, and 3G7; 72 h GyH1-infected MDCC-MSB1 cell lysates reacted with 2B5, 2F2, 3F5, 3H1 and 3G7, respectively. The VP1 protein is 57 kD.
Point 8: Line 152-153: Please also specify positive controls from other viruses that were used. Are these purified proteins as well?
Response 8: We are very sorry for the inconvenience caused to your understanding. Thank you very much for your kind advice. The positive controls we use for other viruses are infected cell lysates, not purified proteins.
Point 9: Line 167-172: What is the detection limit of the qPCR assay? The Ct value of 30 seems to be a low number for a cut-off. Do you have a sense of how many copies Ct=30 is?
Response 9: We are very sorry for the inconvenience caused to your understanding. Thank you very much for your kind advice. The detection limit for qPCR was a Ct value of 30. We tested forty previously screened GyH1-positive standards. Subsequently, we performed statistical analysis and combined this with the circovirus detection limit to obtain this result. The detection limit for qPCR was a Ct value of 30. We tested forty previously screened GyH1-positive standards by qPCR. Subsequently, we performed statistical analysis and combined this with the circovirus detection limit to obtain this result. We demonstrated that this result was reliable by performing tests on clinically suspicious samples and validating the test with multiple assays for suspicious samples. Furthermore, we found that copies for Ct=30 were 3×103.
Point 10: Line 175-185: Please indicate clearly that you are testing for the presence of VP1 protein in chicken sera since usually, sera is used to test for the presence of antibodies against a pathogen.
Response 10: This is a very good question. Your reasoning is correct to a certain extent. Thank you very much for your kind advice. In a previous pre-experiment, we detected GyH1 by PCR in the sera of GyH1-infected chickens. Subsequently, we speculated that the shed viral capsid protein (VP1 protein) might be present in the sera of GyH1-infected chickens. We then performed SDS-PAGE and immunoblotting experiments on sera to confirm this suspicion. The results showed that shed VP1 protein was indeed present in the sera of GyH1-infected chickens.
Point 11: Lines 80-81: Can you specify how the chickens were selected? Were the serum samples collected from apparently healthy flocks or those with clinical disease?
Response 11: We are very sorry for the inconvenience caused to your understanding. Thank you very much for your kind advice. We selected apparently healthy chickens based on random sampling. Serum samples were collected from apparently healthy chickens. The chickens previously mentioned as having clinical symptoms were observed five days after the detection and were not found at the time of observation.
Point 12: Line 113: The antiserum referred here is not monoclonal yet.
Response 12: We appreciate your kind words of caution. We have removed the "monoclonal" from line 113.
Point 13: Line 129: Please define CBS.
Response 13: We appreciate your kind words of caution. We have defined CBS in line 129.
Point 14: Line 131: Please specify what the positive was. Is it the purified VP1 protein?
Response 14: We are very sorry for the inconvenience caused to your understanding. Thank you very much for your kind advice. The positive controls we use for other viruses are infected cell lysates, not purified proteins.
Point 15: Line 235: “infestation” to “infection”.
Response 15: We appreciate your kind words of caution. We have amended "infestation" to "infection".
Point 16: Line 245-247, Figure 1A: Please indicate the statistical test used to compare prevalence. If not done, please drop the adjective “statistical” since its use must be exclusively for findings supported by statistical testing.
Response 16: Thank you very much for your kind advice. We used SPSS software (version 25.0; IBM, USA) for statistical testing to compare prevalence. We used SPSS software for statistical testing to compare popularity. We have supplemented the statistical tests with relevant content under Figure 1 and 2.
Point 17: Line 268: “Transmmition” to “transmition”.
Response 17: We appreciate your kind words of caution. We have changed “Transmmition” to “transmition”.

Round 2
Reviewer 3 Report
Thank you for your very thorough response to my comments